# SHINE: SubHypergraph Inductive Neural nEtwork

**Yuan Luo**
Feinberg School of Medicine
Northwestern University
Chicago, IL 60611
`yuan.luo@northwestern.edu`

## Abstract

Hypergraph neural networks can model multi-way connections among nodes of the graphs, which are common in real-world applications such as genetic medicine. In particular, genetic pathways or gene sets encode molecular functions driven by multiple genes, naturally represented as hyperedges. Thus, hypergraph-guided embedding can capture functional relations in learned representations. Existing hypergraph neural network models often focus on node-level or graph-level inference. There is an unmet need in learning powerful representations of subgraphs of hypergraphs in real-world applications. For example, a cancer patient can be viewed as a subgraph of genes harboring mutations in the patient, while all the genes are connected by hyperedges that correspond to pathways representing specific molecular functions. For accurate inductive subgraph prediction, we propose SubHypergraph Inductive Neural nEtwork (SHINE). SHINE uses informative genetic pathways that encode molecular functions as hyperedges to connect genes as nodes. SHINE jointly optimizes the objectives of end-to-end subgraph classification and hypergraph nodes' similarity regularization. SHINE simultaneously learns representations for both genes and pathways using strongly dual attention message passing. The learned representations are aggregated via a subgraph attention layer and used to train a multilayer perceptron for inductive subgraph inferencing. We evaluated SHINE against a wide array of state-of-the-art (hyper)graph neural networks, XGBoost, NMF and polygenic risk score models, using large scale NGS and curated datasets. SHINE outperformed all comparison models significantly, and yielded interpretable disease models with functional insights.

## 1 Introduction

Hypergraph neural networks have recently emerged as a series of successful methods to model multi-way connections that are beyond pairwise associations among nodes of the graphs. Multi-way connections are common in many real-world applications and, in particular, genetic medicine. From genetic medicine's perspective, pathways or broadly speaking gene sets encode the relationship among multiple genes that collectively correspond to a molecular function [1], which can be used in machine learning models to account for disease mechanisms more intuitively and accurately than individual genes [2]. Genetic pathways or gene sets encode functional relations among multiple genes (see Appendix for detailed explanations), which can be naturally modeled as hyperedges connecting all involved nodes (e.g., genes). Thus, hypergraph-guided embedding can capture functional relations in learned representations.

Existing hypergraph neural network models often adopt the semi-supervised learning (SSL) paradigm to assign labels to initially unlabeled nodes in a hypergraph [3–5]. Other methods have focused on learning graph representations [6, 7]. Node-level and graph-level representations give either local or overarching views of the graphs, i.e., at the two extremes of hypergraph topological structures.

There is an unmet need in learning powerful representations of subgraphs in hypergraphs. Such capabilities are important in real-world applications such as genetic medicine. For example, cancer patients can be viewed as subgraphs of genes that harbor mutations, while all the genes are connected by hyperedges that correspond to pathways or gene sets representing specific molecular functions. Powerful subgraph representations will lead to the capability to more accurately account for the patient's pathophysiology. For regular graphs where edges connect node pairs, several subgraph representation learning algorithms were proposed, including methods that can use the learned representations to make predictions for subgraphs with fixed sizes [8] or varying sizes [9]. There are currently few if any work on inductive inference for varying-sized subhypergraphs. In this work, we propose a new framework named SHINE: SubHypergraph Inductive Neural nEtwork. We share our source code at https://github.com/luoyuanlab/SHINE. Our contributions are as follows:

- To the best of our knowledge, SHINE is the first model to effectively learn subgraph representations for hypergraphs, use the learned representations (for seen subgraphs) and inductively infer representations (for unseen subgraphs) for downstream subgraph predictions.
- Novel applications in the field of genetic medicine on Next Generation Sequencing (NGS) datasets across diverse diseases show significant performance improvements by SHINE over a wide array of state-of-the-art baselines.
- In addition to learning and inductively inferring subgraph representations, SHINE simultaneously learns the representations of nodes and hyperedges. This brings interpretation advantages, allowing assessing pathways (hyperedges) correlations and reasoning on multiple molecular functions mutually interacting and collectively contributing to disease onset and progression.

## 2   Related Work

**Graph Neural Networks.** Graph representation learning maps graphs or their components to vector representations and has attracted growing attention over the past decade. Recently, graph neural networks (GNNs), which can learn a distributed representation for a graph or a node in a graph, are widely applied to a variety of areas including computer vision and image processing [10, 11], molecular structure inference [12, 13], natural language processing [14–16], and healthcare [17, 18]. GNN recursively updates the representation of a node in a graph by aggregating the feature vectors of its neighbors and itself, e.g. [19]. The graph-level representations can then be obtained through set pooling (e.g., [20]) or graph coarsening (e.g., [21]) to aggregate the node representations in the graph. The reader is referred to a comprehensive book [22] on the topic of graph neural networks.

**Hypergraph neural network.** Hypergraph neural networks [6, 4, 3, 7] have become a popular approach for learning on multi-way relations from data. Early work on hypergraph learning, e.g., [23], formulated hypergraph message passing using spectral theory of hypergraphs. This formulation and its variants [3, 4, 24–27] essentially adopted clique expansion to extend graph convolutional network (GCN) for hypergraph learning. Others methods applied attention mechanism to aggregate the information across the hypergraph [6, 7] or directly learned node representations to preserve the proximity of nodes sharing a hyperedge or having similar neighborhoods [28]. In both formulations, messages were passed to the node of interest from its immediate neighbors, and added layers allow propagation of messages to a farther neighborhood. A very recent model implements hypergraph neural network layers in a generalized way as compositions of two multiset functions that are approximated by neural networks [29].

**Subgraph representation learning and prediction.** Recent studies on subgraph embeddings and prediction starts with learning representations of small subgraphs. [8] encoded small fixed-sized subgraphs for subgraph evolution prediction. SubGNN [9] learned representations for varying-sized subgraphs through neighborhood, position and structure channels using random patches distributed throughout the graph. [30, 31] learned subgraph representations by pooling local structures to aid the predictions of the entire graphs. Note modeling hyperedges as another type of nodes turns hypergraphs into bipartite (and heterogeneous) graphs, making SubGNN a potential baseline for subhypergraph inferencing. On the other hand, most of the existing general heterogeneous graph neural network models do not support subgraph inferencing [32–36].

The intersection of hypergraph neural network and subgraph representation learning is currently underexplored. While the above methods focus on either hypergraph learning or subgraph learning,

Table 1: Common notations used throughout the paper.

| Symbol | Definition | Symbol | Definition |
|---|---|---|---|
| $\mathcal{H}$ | An undirected hypergraph | $\mathcal{N}, |\mathcal{N}|$ | Set of hypergraph nodes and size |
| $\mathcal{E}, |\mathcal{E}|$ | Set of hypergraph hyperedges and size | $d$ | Hidden layer size |
| $\mathbf{H}$ | Hyperedge incidence matrix | $\circ$ | Operation composition |
| $n$ | Number of subgraphs | $*$ | Element-wise multiplication |

none of the methods consider subgraph prediction for hypergraphs. Technically, subgraphs can be viewed as a hyperedge and studies on link prediction could predict the existence of a hyperedge [37]. However, few if any such studies addressed the problems of differentiating the classes of the subgraphs, which is especially important in genetic medicine where subgraphs and hyperedges have different real-world meanings. For example, a hyperedge corresponds to a genetic pathway from curated knowledge and a subgraph corresponds to a patient with mutated genes as its nodes.

To sum, there is a major unmet need regarding varying-sized subgraph inference for hypergraphs, and even more so in the inductive learning setting. Our proposed framework SHINE provides an end-to-end framework that operates on hypergraphs and performs inductive subgraph inferencing.

## 3 Methods

We first outline the workflow of SHINE, see Table 1 for symbol definitions. We develop a strongly dual attention message passing algorithm to propagate information between nodes and hyperedges, and across layers. We develop a weighted subgraph attention mechanism to learn the subgraph representation by integrating representations of hypergraph nodes. We next explain each step.

### 3.1 Collecting Genetic Pathways

We use the Molecular Signatures Database (MSigDB) [1] and focus on MSigDB's curated pathway (gene set) collection, which contains human gene sets that are canonical representations of a biological process compiled by domain experts. There are 21,587 genes in MSigDB Pathways. Some pathways may overlap with others and have been filtered by MSigDB to remove interset redundancy. Genes with unknown functions are not included in the pathways and not used for classification, as our focus here is on interpretable modeling through inferencing with known molecular functions. Adding genes with unknown functions to study their impact will be our future work.

### 3.2 Hypergraph Learning

We first review the hypergraph analysis basics. Different from a simple graph, a hyperedge in a hypergraph connects two or more vertices. A hypergraph is defined as $\mathcal{H} = (\mathcal{N}, \mathcal{E})$, which includes a set of nodes $\mathcal{N}$, a set of hyperedges $\mathcal{E}$. In the case of genetic medicine, we model genes as hypergraph nodes, i.e., $\mathcal{N} = \{g_1, ..., g_{|\mathcal{N}|}\}$, and pathways as hyperedges, i.e., $\mathcal{E} = \{p_1, ..., p_{|\mathcal{E}|}\}$, where $|\mathcal{N}|, |\mathcal{E}|$ are sizes of the nodes and hyperedges respectively. The hypergraph's topological structure can be denoted by an $|\mathcal{N}| \times |\mathcal{E}|$ incidence matrix $\mathbf{H}$, whose entries are defined as

$$\mathbf{H}_{ij} = \begin{cases} 1 & \text{if node } g_i \in \text{ hyperedge } p_j \\ 0 & \text{if node } g_i \notin \text{ hyperedge } p_j \end{cases} \tag{1}$$

Generally speaking, each node in the hypergraph may be accompanied by a $d$-dimensional node feature/embedding matrix $\mathbf{N} \in R^{|\mathcal{N}| \times d}$, where each row corresponds to a node's feature/embedding. The hypergraph with its topological structure and node features can be represented succinctly as $\mathcal{H} = (\mathbf{H}, \mathbf{N})$.

Fig. 1 (a) shows a schematic of the constructed genome hypergraph with nodes denoted by circles and hyperedges denoted by colored arcs. While a pathway can contain multiple genes, a gene can also contribute to multiple pathways. That is, we can have multiple hyperedges incident on the same node (gene), as can be seen in Fig. 1 (a) nodes $n_2, n_5, n_{10}$.

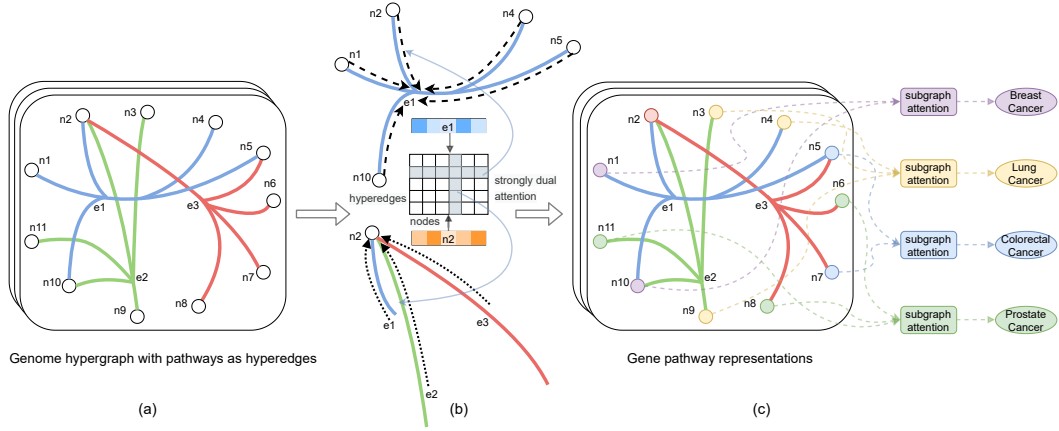

Figure 1: SHINE's strongly dual attention mechanism for message passing for the genome hypergraph, and its use of subgraph attention to integrate gene nodes in the feature learning for subgraphs.

## 3.3 Strongly Dual Attention Message Passing

**Hyperedge attention over nodes.** For a hyperedge $p_j \in \mathcal{E}$, in order to update its hidden representation at layer $k$, we aggregate the information from its incident nodes using the following attention mechanism. We first calculate the hyperedge attention over nodes as in

$$a_E(p_j, g_i) = \exp\left(\mathbf{c}^T \mathbf{s}(p_j, g_i)\right) \bigg/ \left(\sum_{g_{i'} \in p_j} \exp\left(\mathbf{c}^T \mathbf{s}(p_j, g_{i'})\right)\right) \tag{2}$$

where $\mathbf{c}$ is a trainable context vector and the attention ready state $\mathbf{s}(p_j, g_i)$ for a hyperedge-node pair $(p_j, g_i)$ is calculated from the $(k-1)$th layer as in

$$\mathbf{s}(p_j, g_i) = \text{LeakyReLU}\left(\left(\mathbf{W}_N \mathbf{h}_N^{k-1}(g_i) + \mathbf{b}_N\right) * \left(\mathbf{W}_E \mathbf{h}_E^{k-1}(p_j) + \mathbf{b}_E\right)\right) \tag{3}$$

where $*$ denotes element-wise product, $\mathbf{W}_N$ and $\mathbf{b}_N$ ($\mathbf{W}_E$ and $\mathbf{b}_E$) are the transformation weights and bias of the nodes (the hyperedges) for the attention ready state. This is motivated by the observation that different nodes (genes) contribute differently to the hyperedges (pathways), thus we need proper attentions across the nodes to up- or down-weight their contributions when aggregating their representations to compute the representation of the hyperedge. Once we have the hyperedge attention over nodes, we calculate the hyperedge's representation in layer $k$ from the nodes' representations in layer $k-1$ (equation 4) where $\sigma$ is the nonlinearity layer (ReLU in our experiment).

$$\mathbf{h}_E^k(p_j) = \sigma\left(\sum_{g_i \in p_j} a_E(p_j, g_i)\, \mathbf{h}_N^{k-1}(g_i)\right) \tag{4}$$

**Node attention over hyperedges.** For a node $g_i \in \mathcal{N}$, in order to update its hidden representation at layer $k$, we aggregate the information from its incident hyperedges using the following attention mechanism. We first calculate the node attention over hyperedges as in

$$a_N(g_i, p_j) = \exp\left(\mathbf{c}^T \mathbf{s}(p_j, g_i)\right) \bigg/ \left(\sum_{p_{j'} \ni g_i} \exp\left(\mathbf{c}^T \mathbf{s}(p_{j'}, g_i)\right)\right) \tag{5}$$

where $\mathbf{c}$ is the same trainable context vector as used in hyperedge attention calculation and the attention ready state $\mathbf{s}(p_j, g_i)$ for hyperedge-node pair $(p_j, g_i)$ is calculated as in equation 3. This allows us to have the node's attention over the hyperedges, i.e., we can weight hyperedges' contributions when aggregating their representations to compute the representation of the node. We calculate the node's representation in layer $k$ from the hyperedges' representations in layer $k-1$ as in

$$\mathbf{h}_N^k(g_i) = \sigma\left(\sum_{p_j \ni g_i} a_N(g_i, p_j)\, \mathbf{h}_E^{k-1}(p_j)\right) \tag{6}$$

Note that different from HyperGAT, here the calculation of hyperedge's and node's attentions share the same underlying dual-attention matrix as shown in Fig. 1 (b), which is essentially unstandardized covariance matrix. Such parameter sharing across hyperedges and nodes allows us to cross-regulate

the learning of their mutual attentions to prevent overfitting. This difference not only allows for simplification of the model, but also is more consistent with the notion of duality. The dual $\mathcal{H}^*$ of the hypergraph $\mathcal{H}$ is a hypergraph with $\mathcal{H}$'s vertices and edges interchanged, and we should have $(\mathcal{H}^*)^* = \mathcal{H}$. It is easily provable that the dual-attentions for $(\mathcal{H}^*)^*$ are the same as those for $\mathcal{H}$. Such a self-dual statement is generally not true for the attentions proposed in HyperGAT due to their unsymmetrical way of calculating the node-level and the edge-level attentions, despite that the HyperGAT attention was termed as "dual" attention. For this reason, we term our attention message passing scheme as strongly dual attention message passing.

**Hypergraph regularization.** One important intuition about graph and hypergraph convolutional network is that the learned representations for nodes with similar context of (hyper)edges should be similar. In the case of a simple graph $G = (V, E)$, this is to minimize the summed distance $\sum_{(u,v) \in E} \|h_u - h_v\|^2$ or its weighted variants. Instead of using it as an explicit regularizer, graph or hypergraph convolutional networks leverage an appropriately defined graph or hypergraph Laplacian. As noted in [23], the hypergraph Laplacian is $\mathbf{\Delta} = \mathbf{I} - \mathbf{\Theta}$ where $\mathbf{I}$ is the identity matrix and $\mathbf{\Theta}$ is defined as (let $\mathbf{W}$ be a diagonal matrix with diagonal entries as hyperedge weights)

$$\mathbf{\Theta} = \mathbf{D}_v^{-1/2} \mathbf{H} \mathbf{W} \mathbf{D}_e^{-1} \mathbf{H}^T \mathbf{D}_v^{-1/2} \tag{7}$$

Here, different from hypergraph convolutional networks, we use explicit regularization on the similarity of representations of nodes with similar hyperedge context. Let $\mathbf{X}$ be the matrix of the learned nodes' representations, where row $\mathbf{X}_i = \mathbf{h}_N^K(g_i)$ and the $K$th layer is the last hypergraph message passing layer. We can define the regularizer as

$$\mathcal{L}_{reg} = \sum_{i,j} \left( (\mathbf{X}_i \mathbf{X}_i^T - 2\mathbf{X}_i \mathbf{X}_j^T + \mathbf{X}_j \mathbf{X}_j^T) * \mathbf{\Theta}_{ij} \right) \tag{8}$$

Intuitively, the more hyperedges are incident on the node pair $i, j$, the more we should penalize their representational differences. On the other hand, the regularizer down-weights the penalization if a hyperedge connects many nodes or if a node has many incident hyperedges, indicating lack of specificity for hyperedges and nodes, respectively.

## 3.4 Weighted Subgraph Attention

The multiple layers of strongly dual attention message passing allow learning the nodes' and hyperedges' representations. However, the instance for the classification algorithm is a subgraph (e.g., a patient, who has mutations in multiple genes (nodes)). From the hypergraph perspective, a patient $j(1 \leq j \leq n)$ can be considered as a subhypergraph $\mathcal{G}_j$ whose nodes (genes) have mutations in $j$ and are a subset of those of $\mathcal{H}$. This is shown in Fig. 1 (c) where different node colors in the hypergraph denote different patients. In order to calculate the subgraph's representation from its component nodes' representations at the $K$th layer, we use the following weighted subgraph attention (WSA) mechanism, inspired by [38]. In fact, none of the previous hypergraph methods support subgraph inferencing, and we had to add our WSA module to those models for subgraph inferencing as well. We first compute the subgraph attention over nodes (e.g., $g_i$'s) as in

$$a(\mathcal{G}_j, g_i) = \exp \left( \mathbf{M}_{ji} \mathbf{b}^T \mathbf{h}_N^K(g_i) \right) \Big/ \left( \sum_{g_{i'} \in \mathcal{G}_j} \exp \left( \mathbf{M}_{ji'} \mathbf{b}^T \mathbf{h}_N^K(g_{i'}) \right) \right) \tag{9}$$

where $\mathbf{b}$ is a trainable context vector, $\mathbf{M}$ is the mutation rate feature matrix with each row corresponding to a patient and each column corresponding to a gene. Thus, equation 9 is a mutation rate weighted subgraph attention mechanism. This choice conforms to the intuition that the rate of a mutation is more informative than a categorical indicator of the mutation's occurrence. With these subgraph level attentions, we compute the patient (subgraph) representation from the $K$th layer's gene representations as in

$$\mathbf{h}(\mathcal{G}_j) = \sigma \left( \sum_{g_i \in \mathcal{G}_j} a(\mathcal{G}_j, g_i) \, \mathbf{h}_N^K(g_i) \right) \tag{10}$$

We then stacked the learned patient representations to form the new patient feature matrix, as in

$$\mathbf{S} = [\mathbf{h}(\mathcal{G}_1)^T \mid \mathbf{h}(\mathcal{G}_2)^T \mid ... \mid \mathbf{h}(\mathcal{G}_n)^T]^T \tag{11}$$

where each row is a patient (subgraph) embedding.

## 3.5 Inductive Classification on Subgraphs

Let the learned feature matrix be $\mathbf{S}$ and feed it into a softmax classifier

$$\mathbf{Z} = \text{softmax}(\mathbf{W}^{(1)} \, (ReLU \circ FC)^{(2)}(\mathbf{S}) \, + \, \mathbf{W}^{(0)}) \tag{12}$$

where (2) in the superscript indicates two MLP layers ($FC$=Fully Connected layer). The loss function is defined as the cross-entropy error over all subjects in all classes as in

$$\mathcal{L} = -\sum_{j \in \mathcal{Y}_D} \sum_{f=1}^{F} \mathbf{Y}_{jf} \ln \mathbf{Z}_{jf} + \mathcal{L}_{reg} \tag{13}$$

where $\mathcal{Y}_D$ is the training set of subjects that have labels and $F$ is the dimension of the output labels, which is equal to the number of classes. $\mathbf{Y}$ is the label indicator matrix. Note that the subgraph attention layer allows us to compute any patient's representation, which effectively eliminates the need for access to test set patient features during training, making the model inductive. Existing models such as HyperGCN and HGNN are transductive, we cascade our subgraph attention layer on top of these models and make them inductive to serve as our comparison models.

## 4 Experiments

We conducted experiments on real-world datasets in genetic medicine. Both datasets have more than 20 different classes, indicating significant complexity of the prediction tasks. These datasets are different in nature, e.g., curated from literature vs. obtained directly from high-throughput sequencing, and multi-class vs. multi-class multi-label. Our experiments are motivated by the fact that massive genomic data call for novel methods and present unique technical challenges, in this case inductive subgraph inferencing on hypergraph. The summary statistics for each dataset is shown in Table 2, and the description of each dataset is as follows. Most pathways have small to medium sizes, see Table 2 pathway sizes IQR. In fact, even at the 95th percentile, the pathway size is just over 200. On the other hand, we observed that the larger the pathway (hyperedge), the more subgraph it is incident on, and the less attention our model will give it as a discriminative feature. The DisGeNet and the TCGA-MC3 datasets are publicly available[1], and this study is approved by Northwestern University Institutional Review Board.

### 4.1 Disease Type Prediction with DisGeNet Data

In this experiment, we have used the DisGeNet dataset [39] that is a collection of mutated genes involved in human diseases compiled from expert curated repositories, GWAS catalogs, animal models and the scientific literature. In the following text, we abuse terminology to use "gene" to really mean "variants in the gene". We model genes as hypergraph nodes and diseases as hyperedges. Each disease is labeled with one or more of 22 MeSH codes, and the task is a multi-class multi-label classification problem. We used 6:2:2 train:validation:test partition, and the split distribution is shown in Appendix. The DisGeNet dataset has 6226 pathways and 9133 genes involved in 8383 diseases.

### 4.2 Cancer Type Prediction with NGS Somatic Mutations Data

In this experiment, we have used the consensus somatic mutations for TCGA subjects produced by the Multi-Center Mutation Calling in Multiple Cancers (MC3) project [40]. Aiming to enable robust cross-tumor-type analyses, the MC3 approach applied an ensemble of 7 mutation-calling algorithms and assigned a PASS identifier to a mutation that was called by 2 or more variant callers out of the total 7 callers [40]. The MC3 approach accounted for variance and batch effects introduced by the rapid advancement of DNA extraction, hybridization-capture, and sequencing over time. Following this approach, we restricted our analysis to PASS calls in order to maintain sample sizes and uniformity in mutation calling. Each subject is labeled with one of 25 cancer types, and the task is a multi-class classification problem. We used 6:2:2 train:validation:test partition, stratified by cancer types, and the split distribution is shown in Appendix. The TCGA-MC3 dataset has 6229 pathways and 18059 genes involved in 9012 subjects in total.

---

[1]DisGeNet: https://www.disgenet.org/; TCGA-MC3: https://gdc.cancer.gov/about-data/publications/mc3-2017

Table 2: Real-world hypergraph datasets for subgraph inference used in our work. For hyperedge sizes, shown are medians and interquartile ranges. From the distribution, it is clear that the hyperedge size has a positive skewness.

| Dataset | # hypernodes | # hyperedges | Hyperedge size | # classes | # subgraphs |
|---------|--------------|--------------|----------------|-----------|-------------|
| DisGeNet | 9133 | 6226 | 25 (12 - 57) | 22 | 8383 |
| TCGA-MC3 | 18059 | 6229 | 33 (15 - 77) | 25 | 9012 |

## 4.3 Baselines

We compared SHINE with the following state-of-the-art baselines, we use validation datasets to tune parameters and hyperparameters, please see Appendix for details.

- Hypergraph neural networks (**HGNN**) [3] uses clique expansion to transform the hypergraph to graph and uses Chebyshev approximation to derive a simplified hypergraph convolution operation.

- **HyperGCN** [4] represents a hyperedge by a selected pairwise simple edge connecting two most unlike nodes, and adds the remaining nodes in the hyperedge as mediators.

- **HyperGAT** [7] learns node representations by aggregating information from nodes to edges and vice versa. Different from SHINE, HyperGAT uses alternating attention instead of strictly dual attention and has no regularization on nodes with similar context of hyperedges.

- **AllSetTransformer** and **AllDeepSets** are two variants (attention-based and MLP-based respectively) of set-based methods (e.g., compositions of two multiset functions that are permutation invariant on their input multisets) for exploiting hyperedges in hypergraphs [29].

- **SubGNN** [9] was applied to the hypergraph by viewing the nodes and hyperedges as two types of vertices of a bipartite graph [41].

- The multilayer perceptron (**MLP**) baseline evaluates how a simple feed-forward neural network with hypergraph regularization and subgraph attention performs, by replacing dual attention with MLP.

- Polygenic risk score (**PRS**) is currently a widely used standard practice in genetic medicine and calculates disease risk from genotype profile using regression [42].

- Non-negative matrix factorization (**NMF**) discovers low-dimensional structure from high-dimensional multi-omic data and enables inference of complex biological processes [43].

- **XGBoost** is an end-to-end tree boosting system and a state-of-the-art machine learning method [44] that frequently achieves the top results on many machine learning challenges.

To assess whether performance changes are due to added information (e.g., pathway information) and/or better utilization of the added information, we run PRS, NMF, and XGBoost in the following three settings: gene features only, pathway features only, and both gene and pathway features.

## 5 Results

The held-out test set micro-averaged F1 scores (micro-F1) for our proposed method SHINE and all the other comparison models are in Table 3. Comparing all the models, we can see that SHINE clearly outperforms a comprehensive array of state-of-the-art baselines in various configurations, with non-overlapping standard deviation intervals. PRS is indeed a competitive baseline, as can be seen from its close performance compared with XGBoost that frequently topped many machine learning challenges' leaderboards. Previously state-of-the-art hypergraph neural network models (HyperGCN, HGNN, HyperGAT) do not always outperform PRS and XGBoost (e.g., on the TCGA-MC3 dataset). On the other hand, pathway as features could improve performance if used properly, whether alone or jointly with genes. This comparison shows that genetic pathway information is useful to disease type classification, consistent to the intuition that pathways encode molecular functional mechanisms that underlie the disease etiology. However, properly utilizing such information is non-trivial, as evidenced by the difficulty to outperform PRS and XGBoost models by NMF models and hypergraph models including HyperGCN, HGNN and HyperGAT. Given that difficulty, SHINE

Table 3: Held-out test set micro-F1 on real-world datasets. Standard deviations are provided from runs with 10 random seeds. SHINE significantly outperforms all the state-of-the-art comparison models. PRS: polygenic risk score. NMF: non-negative matrix factorization. Best model in bold.

| Model _Metrics_ | Feature | DisGeNet Dataset Test Micro F1 | TCGA-MC3 Dataset Test Micro F1 |
|---|---|---|---|
| PRS | gene | 0.6303 | 0.4981 |
| PRS | pathway | 0.6461 | 0.5047 |
| PRS | gene+pathway | 0.6512 | 0.5042 |
| XGBoost | gene | $0.6259 \pm 0.0012$ | $0.4927 \pm 0.0058$ |
| XGBoost | pathway | $0.6467 \pm 0.0035$ | $0.4936 \pm 0.0092$ |
| XGBoost | gene+pathway | $0.6486 \pm 0.0036$ | $0.5117 \pm 0.0084$ |
| NMF | gene | $0.6167 \pm 0.0040$ | $0.4181 \pm 0.0125$ |
| NMF | pathway | $0.5867 \pm 0.0039$ | $0.4842 \pm 0.0057$ |
| NMF | gene+pathway | $0.5847 \pm 0.0045$ | $0.4839 \pm 0.0032$ |
| SubGNN (bipartite) | gene+pathway | $0.6137 \pm 0.0097$ | $0.4025 \pm 0.0049$ |
| HyperGCN | gene+pathway | $0.6638 \pm 0.0028$ | $0.4384 \pm 0.0095$ |
| HGNN | gene+pathway | $0.6809 \pm 0.0027$ | $0.4504 \pm 0.0042$ |
| HyperGAT | gene+pathway | $0.6495 \pm 0.0050$ | $0.4721 \pm 0.0032$ |
| MLP | gene+pathway | $0.6331 \pm 0.0056$ | $0.4249 \pm 0.0165$ |
| AllDeepSets | gene+pathway | $0.6309 \pm 0.0147$ | $0.4324 \pm 0.0220$ |
| AllSetTransformer | gene+pathway | $0.6355 \pm 0.0160$ | $0.4904 \pm 0.0158$ |
| SHINE | gene+pathway | $\mathbf{0.6955 \pm 0.0034}$ | $\mathbf{0.5319 \pm 0.0049}$ |

still attained the best performance on each dataset. Intuitively speaking, HyperGCN and HGNN focus on similarity regularization: hypergraph nodes with similar context of hyperedges should have similar representations. HyperGAT's attention mechanism gears more towards minimizing the classification loss. SHINE, in an attempt to balance the similarity regularization with the end-to-end classification task via strongly dual attention mechanism, achieved better trade-off between the two objectives and effectively integrated the functional pathway's (hyperedge) information with individual gene's (node) information.

The importance of the strong duality follows naturally from that SHINE outperforms SubGNN (bipartite) with non-overlapping standard deviation intervals and wide separation. In addition, although MLP has been frequently used to approximate a target function, in the setting of large hypergraph (e.g., both hypergraphs have a few thousand-nodes hyperedges), it can still be quite challenging to approximate an ideal target function and explicit dual attention formulation wins out. The results from both AllDeepSets and AllSetTransformer have non-overlapping standard deviation intervals, in fact wide separation, with their counterparts from SHINE. These results echo with our observations that strongly dual attention explores the hypergraph propagation from a different angle than both AllDeepSets and AllSetTransformer, and suggest that effectively combining both angles could be an interesting future direction. Also note that in general, we have some performance drop when moving from the DisGeNet dataset to the TCGA-MC3 dataset, likely due to the fact that the former uses genetic features from curated literatures and the latter is from high throughput sequencing intended for data-driven discovery. Both complex classification tasks (>20 classes) are uniquely challenging because diseases may have overlapping disrupted molecular functions (genetic pathways, hyperedges), especially for the TCGA-MC3 experiment that is distinguishing subcategories of similar diseases as they are loosely all cancers. In addition, both tasks exhibit class distributional shift between the train and the test datasets, as shown in Appendix Tables 1 and 2, and have been designed to require inductive inference on subgraphs with highly variable hyperedge sizes. Strong performance of SHINE on these tasks thus suggests that our model can leverage its relational inductive biases for more robust generalization. Ablation studies further confirmed the contributions from each of the key components, including strictly dual attention massage passing, weighted subgraph attention and hypergraph regularization (see Appendix for details).

**Model interpretation.** SHINE simultaneously learns the representations of nodes and hyperedges, which are then used to learn and inductively infer subgraph representations. This brings some interpretation advantages as it allows assessing pathways (hyperedges) correlations and reasoning multiple molecular functions mutually interacting and collectively contributing to the disease onset and progression. In addition, SHINE has built-in measures to prevent or discourage genes belonging

Table 4: Top enriched genetic pathways associated with different cancer risks. The text color indicates the source database for pathways that MSigDB integrated: BioCarta, Reactome, WikiPathways, Pathway Interaction Database, KEGG.

| BRCA | LUAD | LGG | HNSC |
|---|---|---|---|
| Stress pathway | PTK6 stabilizes HIF1$\alpha$ | Citrate cycle TCA cycle | Apoptotic factor response |
| 4-1BB pathway | ErbB3 pathway | Cytosine methylation | Programmed cell death |
| VIP pathway | Hypertrophic cardiomyopathy | TCA cycle and deficiency of pyruvate dehydrogenase | MECP2 regulates neuronal receptors and channels |
| CD40 pathway | Diseases of metabolism | Glutathione metabolism | FRA pathway |
| TOLL pathway | TFAP2 regulates growth factors transcription | Digestion of dietary carbohydrate | Caspase activation via extrinsic apoptotic signalling |

to the same functional class (e.g., promoting immune reactions) from having drastically different representations (e.g., opposite directions), a phenomenon that will pose interpretation difficulty to other models that do not employ SHINE's hypergraph regularization. We identify the top pathways that are enriched in different cancers using the attention weights learned for SHINE, as shown in Table 4. From the table, we see that many of the listed pathways reflect innate key events in the development of individual or multiple types of cancers, consistent with genetic and medical knowledge from wet lab (e.g., TNF/Stress Related Signaling [45]). We showcase interpretations for breast cancer and lung cancer here, and refer the reader to the Appendix for full interpretation of Table 4. For breast cancer, TNF$\alpha$ is not only closely involved in its onset, progression and in metastasis formation, but also linked to therapy resistance [45]. Regarding the 4-1BB pathway, studies have suggested HER2/4-1BB bispecific molecule as a candidate of alternative therapeutic strategy to patients in HER2-positive breast cancer [46]. VIP/PACAP and their receptors have prominent roles in transactivation of the Epidermal growth factor (EGF) family and growth effects in breast cancer [47]. For lung cancer, the ErbB3 receptor recycling controlled by neuroregulin receptor degradation protein-1 is linked to lung cancer and small inhibitory RNA (siRNA) to ErbB3 shows promise as a therapeutic approach to treatment of lung adenocarcinoma [48]. Lung cancer is also modulated by multiple miRNAs interacting with the TFAP2 family [49].

## 6 Discussion, Limitation and Future Work

In addition to being significantly more accurate and interpretable, SHINE uses inductive subgraph inferencing that works well with minibatch, and scales well to large scale problems, as showcased by real-world experiments. It is known that GNN suffers from over-smoothing when the number of layers increases, as increasingly globally uniform representation of nodes may be developed. On the other hand, attention could limit this phenomenon by limiting to a restricted set of nodes. The effect of hypergraph regularization, while also smoothing, happens on a local scale as part of a direct optimization objective and does not accumulate with increasing number of layers. Such decoupling between attention and local smoothing allows SHINE to better explore the optimization landscape.

Our work has limitations. We assumed that the hyperedges are known in advance. However, in reality, as our domain knowledge increases and evolves, we need to account for unknown hyperedges and, better, simultaneously discover novel hyperedges from data while predicting disease classes. Such a task has important clinical utilities in genetic medicine to discover new genetic pathways that may underlie disease etiology, and will be our future work. Moreover, strongly dual attention explores the hypergraph propagation from a different angle than both AllDeepSets and AllSetTransformer, and effectively combining both angles could be an interesting future direction. Another line of future work is to derive a hypergraph coarsening model on top of SHINE. SHINE currently has flat hypergraph layout and does not learn hierarchical representations of hypergraphs. The emerging technique of spatial transcriptomics can enable discovery of localized and hierarchical gene expression patterns [50, 51]. A flexible hypergraph coarsening model that can effectively learn hierarchical network structure out of the hypergraphs can shed light on the organizations of the hyperedges (e.g., pathways representing synergistic molecular functions in certain tissue context).

From the application point of view, detecting tumor subtypes is often interesting, and we expect to extend our method to such detections using multi-modal data when large shared datasets will become available [52]. To certain extent, the TCGA labels we used reflect subtypes of organ-specific primary tumors, e.g., LUAD vs. LUSC in lung cancer, KIRC vs. KIRP in kidney cancer. On the other hand, identifying drivers genes and pathways for cancer types and other disease subtypes continue to be biologically important [53] and will be increasingly fruitful with simultaneously collected deep genetic and phenotypic data on the same patients [54, 55].

The field of genetic medicine encompasses areas of molecular biology and clinical phenotyping to explore new relationships between disease susceptibility and human genetics. Though appearing as a single field, it revolutionizes the practice of medicine in preventing, modifying and treating many diseases such as cardiovascular disease and cancer [56]. We expect SHINE to be a useful tool in the quest of broadly advancing the knowledge on disease susceptibility. In these real-world applications, a subject's genetic profile may contain individual characterizing information. Thus, this work should never be used in violation of an individual's privacy, and the necessary steps of IRB review and execution of data user agreement need to be properly completed prior to the study.

## 7    Conclusions

We proposed a novel framework termed SubHypergraph Inductive Neural nEtwork (SHINE) for inductive subgraph inferencing on hypergraphs, designed for jointly optimizing the objectives of end-to-end subgraph classification and similarity regularization for representations of hypergraph nodes with similar context of hyperedges. We showed that SHINE improved the performance (micro-F1) of the learned model for disease type prediction for complex (>20 classes) genetic medicine datasets of different characteristics and under different settings (e.g., multi-class and/or multi-label). Genetic pathways directly correspond to molecular mechanisms and functions, which are more informative than individual genes and are represented as hyperedges in SHINE. The novel formulation of disease classification as a subgraph inferencing problem allows a hypergraph neural network to link correlated pathways, i.e., interacting molecular mechanisms, to disease etiology. This leads to better performance with added interpretability. We compared SHINE with a wide array of state-of-the-art (hyper)graph neural networks, XGBoost, NMF, and PRS models with different configurations of genes and pathways as features. SHINE consistently outperformed all state-of-the-art baselines significantly in each of the disease classification and cancer classification tasks. Feature analysis of the learned pathway groups that are automatically identified by SHINE in a data-driven fashion offered significant clinical insights about multiple molecular mechanisms that interact and are associated with disease types and status.

## Acknowledgments and Disclosure of Funding

This work was supported in part by NIH grants R01LM013337 and U01TR003528.

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
