# A  Appendix for SHINE: SubHypergraph Inductive Neural nEtwork

**Datasets details.**   In this section, we give additional details on the datasets used in this paper. The DisGeNet dataset is a collection of mutated genes involved in human diseases compiled from expert curated repositories, GWAS catalogs, animal models and the scientific literature. Each disease is labeled with one or more of 22 MeSH codes, and the task is a multi-class multi-label classification problem. We used 6:2:2 train:validation:test partition, and the split distribution is shown in Table 1. The DisGeNet dataset has 6226 pathways and 9133 genes involved in 8383 diseases in total. The TCGA-MC3 dataset records somatic mutations for subjects in The Cancer Genome Atlas (TCGA). The genetic variants are stored in a specially formatted file. A row in the file specifies a particular variant (e.g., Single Nucleotide Polymorphism or insertion/deletion), its chromosomal location, and what proportion of the sequencing reads covering that chromosomal location have that variant, among other characteristics. Each subject is labeled with one or more of 25 cancer types, and the task is a multi-class classification problem. We used 6:2:2 train:validation:test partition, stratified by cancer types, and the split distribution is shown in Table 2. The TCGA-MC3 dataset has 6229 pathways and 18059 genes involved in 9012 subjects in total.

**Genetic pathways.**   Genetic pathways are a valuable tool to assist in representing, understanding, and analyzing the complex interactions between molecular functions. The pathways contain multiple genes (can be modeled using hyperedges) and correspond to genetic functions, including regulations, genetic signaling, and metabolic interactions. They have a wide range of applications, including predicting cellular activity and inferring disease types and status [1]. For a simplified and illustrative example, a signaling pathway p1 (having 20 genes) sensing the environment may govern (the governing function embodied as a pathway p2 having 15 genes) the expression of transcription factors in another signaling pathway p3 (having 23 genes), which then controls (the controlling function embodied as a pathway p4 having 34 genes) the expression of proteins that play roles as enzymes in a metabolic pathway p5 (having 57 genes). In general, there will be partial overlap between pathways p1 and p2, p2 and p3, p3 and p4, p4 and p5, and other potential partial overlaps corresponding to partial overlaps between their corresponding hyperedges.

**Genetic variant calling and filtering for TCGA-MC3 dataset.**   The variants are usually of high dimensionality. For example, in the TCGA-MC3 dataset, even after we retain only the variants that received PASS identifiers, there are still around 3 million variants. Thus, we choose to aggregate their counts according to the affected genes to avoid impractically large matrices. We aggregate genetic variant count at gene level and sum up all the alternative allele counts and reference allele counts in a gene. We calculate the mutation rate for a gene $g$ as in equation 1,

$$\mu(g) = \frac{\sum_{v \in g} C_{ALT}(v)}{\sum_{v \in g} C_{ALT}(v) + \sum_{v \in g} C_{REF}(v)} \tag{1}$$

where variant $v$ belongs to the gene $g$, $C_{ALT}(v)$ is the read depth supporting the variant (alternative) allele in tumor sequencing data and $C_{REF}(v)$ is the read depth supporting the reference allele (non-mutated) in tumor sequencing data.

**Parameter and hyperparameter tuning for models.**   For SHINE and other hypergraph methods, the hyperparameter of hidden dimension $d$ is tuned on the validation dataset with choices from 100 to 1000, at increments of 100. Deep neural network models are often randomly initialized, thus we also run initialization 10 times and report the averages and standard deviations. For the comparison hypergraph neural network models, we used the implementations by the original authors. The hyperparameters were tuned on the validation set using choice grids according to respective papers, or when unspecified, from default grids as with our proposed method (learning rate $\in$ $[0.001, 0.002, 0.005]$, weight decay $\in [0.0001, 0.0005]$, dropout rate $\in [0.4, 0.5, 0.6]$). For PRS, the regularization coefficient $C$ is tuned on the validation dataset with choices from a geometric sequence from 0.001 to 1000 at a multiplying ratio of 10. For NMF, the number of factors is tuned on the validation dataset with choices from 100 to 1000, at increments of 100. For XGBoost, we tuned max tree depth (3, 5, 10), the number of estimators (from 100 to 1000, at increments of 100), and min child weight (0.01, 0.1, 1, 10, 100), using the validation set. For models requiring random initialization, we run initializations 10 times with different seeds and report the averages and standard deviations. We varied the number K of layers from 1 to 4, and found that 2 layers to give the best results for SHINE.

Regarding sensitivity to the hidden dimension, in general, the performance is less sensitive to the hidden dimensions when it is sufficiently big ($\geq$300), with <0.05 change in micro-F1 score. Smaller hidden dimensions (100-200) can lead to >0.05 micro-F1 drop, likely due to insufficient representation power. The optimal hidden dimension is 300 for the TCGA-MC3 dataset and 600 for the DisGeNet dataset. The performance also shows <0.05 change in micro-F1 score when varying other hyperparameters including learning rate, weight decay, dropout rate in their respective grids as specified above.

**Computational complexity.** The complexity of SHINE scales as the following factors grow: the numbers of layers and nodes, the number and size of hyperedges, the size of hidden dimensions, and finally the number and size of subhypergraphs. We implement SHINE on PyTorch, and run it on NVIDIA V100 GPUs. We train SHINE for up to 6000 epochs using Adam [2] and stop training if the validation loss does not decrease for 10 consecutive epochs. The TCGA-MC3 dataset's training times are: MLP ~5 min, HyperGCN ~7 min, AllSetTransformer ~20 min, AllDeepSet ~20 min, SHINE ~30 min, HGNN ~30 min, HyperGAT ~30 min, SubGNN >1 day (excluding prebuild time). The DisGeNet dataset's training times are: MLP ~5 min, HyperGCN ~6 min, AllDeepSet ~13 min, HGNN ~15 min, HyperGAT ~15 min, AllSetTransformer ~16 min, SHINE ~20 min, SubGNN >1 day (excluding prebuild time).

**Ablation study.** To investigate the contribution of key components (e.g., the strictly dual attention massage passing, the usage of hypergraph regularization) in the proposed algorithm to the overall method, we performed an ablation analysis. The previous state-of-the-art hypergraph neural network models in fact serve as some of the steps in the ablation. For example, HyperGAT does not have strict dual attention message passing and does not employ hypergraph regularization. HGNN and HyperGCN apply hypergraph convolution instead of attention message passing. HyperGCN, compared to HGNN, applies approximate hypergraph convolution by representing a hyperedge by a selected pairwise simple edge connecting two most unlike nodes, and adding the remaining nodes

Table 1: Statistics of DisGeNet experiment data. The table includes the distribution of the 22 MeSH categories with more than 100 diseases. The dataset is split into a training set, a validation set and a test set according to a 6:2:2 ratio.

| MeSH | Description | Total | Train | Val | Test |
|---|---|---|---|---|---|
| C01 | Infections | 221 | 135 | 45 | 41 |
| C04 | Neoplasms | 1010 | 626 | 190 | 194 |
| C05 | Musculoskeletal Diseases | 1266 | 765 | 239 | 262 |
| C06 | Digestive System Diseases | 430 | 238 | 91 | 101 |
| C07 | Stomatognathic Diseases | 242 | 156 | 50 | 36 |
| C08 | Respiratory Tract Diseases | 235 | 137 | 52 | 46 |
| C09 | Otorhinolaryngologic Diseases | 299 | 188 | 55 | 56 |
| C10 | Nervous System Diseases | 2960 | 1769 | 619 | 572 |
| C11 | Eye Diseases | 756 | 470 | 150 | 136 |
| C12 | Male Urogenital Diseases | 537 | 337 | 102 | 98 |
| C13 | Female Urogenital Diseases and Pregnancy Complications | 640 | 402 | 118 | 120 |
| C14 | Cardiovascular Diseases | 746 | 441 | 147 | 158 |
| C15 | Hemic and Lymphatic Diseases | 624 | 392 | 108 | 124 |
| C16 | Congenital, Hereditary, and Neonatal Diseases and Abnormalities | 3648 | 2168 | 725 | 755 |
| C17 | Skin and Connective Tissue Diseases | 789 | 459 | 142 | 188 |
| C18 | Nutritional and Metabolic Diseases | 1277 | 725 | 271 | 281 |
| C19 | Endocrine System Diseases | 535 | 327 | 107 | 101 |
| C20 | Immune System Diseases | 415 | 249 | 87 | 79 |
| C23 | Pathological Conditions, Signs and Symptoms | 1795 | 1065 | 387 | 343 |
| C25 | Chemically-Induced Disorders | 135 | 80 | 29 | 26 |
| F01 | Behavior and Behavior Mechanisms | 267 | 164 | 62 | 41 |
| F03 | Mental Disorders | 501 | 295 | 123 | 83 |

Table 2: Statistics of TCGA-MC3 experiment data. The table includes the distribution of the 25 cancer types with more than 100 subjects. The dataset is split into a training set, a validation set and a test set according to a 6:2:2 ratio.

| Cancer | Description | Total | Train | Val | Test |
|---|---|---|---|---|---|
| BLCA | Bladder Urothelial Carcinoma | 411 | 247 | 82 | 82 |
| BRCA | Breast invasive carcinoma | 791 | 475 | 158 | 158 |
| CESC | Cervical squamous cell carcinoma and endocervical adenocarcinoma | 289 | 173 | 58 | 58 |
| COAD | Colon adenocarcinoma | 288 | 173 | 57 | 58 |
| ESCA | Esophageal carcinoma | 184 | 110 | 37 | 37 |
| GBM | Glioblastoma multiforme | 309 | 185 | 62 | 62 |
| HNSC | Head and Neck squamous cell carcinoma | 507 | 304 | 102 | 101 |
| KIRC | Kidney renal clear cell carcinoma | 368 | 220 | 74 | 74 |
| KIRP | Kidney renal papillary cell carcinoma | 281 | 169 | 56 | 56 |
| LAML | Acute Myeloid Leukemia | 137 | 83 | 27 | 27 |
| LGG | Brain Lower Grade Glioma | 510 | 306 | 102 | 102 |
| LIHC | Liver hepatocellular carcinoma | 363 | 217 | 73 | 73 |
| LUAD | Lung adenocarcinoma | 512 | 307 | 103 | 102 |
| LUSC | Lung squamous cell carcinoma | 480 | 288 | 96 | 96 |
| OV | Ovarian serous cystadenocarcinoma | 409 | 245 | 82 | 82 |
| PAAD | Pancreatic adenocarcinoma | 175 | 105 | 35 | 35 |
| PCPG | Pheochromocytoma and Paraganglioma | 178 | 107 | 35 | 36 |
| PRAD | Prostate adenocarcinoma | 493 | 295 | 99 | 99 |
| SARC | Sarcoma | 236 | 142 | 47 | 47 |
| SKCM | Skin Cutaneous Melanoma | 466 | 280 | 93 | 93 |
| STAD | Stomach adenocarcinoma | 438 | 262 | 88 | 88 |
| TGCT | Testicular Germ Cell Tumors | 128 | 77 | 25 | 26 |
| THCA | Thyroid carcinoma | 490 | 294 | 98 | 98 |
| THYM | Thymoma | 122 | 74 | 24 | 24 |
| UCEC | Uterine Corpus Endometrial Carcinoma | 447 | 268 | 90 | 89 |

in the hyperedge as mediators. To evaluate the efficacy of the weighted subgraph attention (WSA), we consider a subgraph simply the sum of the nodes (genes) that are of interest (with mutations) for each patient (subgraph). Finally, we added SHINE with no hypergraph regularization to evaluate the regularization effectiveness. The ablation analysis results are shown in Table 3. From the results, it is clear that SHINE's strictly dual attention message passing outperforms HyperGAT without strictly dual attention message passing. We can see that adding hypergraph regularization further improves performance, in fact, with improvement beyond standard deviation intervals of the regularization-ablated model on both datasets. The weighted subgraph attention (WSA) ablation leads to a larger performance drop than hypergraph regularization ablation, which corroborates the importance of the WSA step. We also notice that the performance drop due to WSA ablation on the TCGA-MC3 dataset is larger than that on the DisGeNet dataset. This is consistent with the fact that the TCGA-MC3 dataset has denser hypergraph and larger subgraphs than the DisGeNet dataset. This is also consistent with the fact that differentiating among cancer subtypes is a more complex and nuanced task than differentiating among disease categories. These observations collectively argue for the benefits of weighted subgraph attention over direct aggregation such as sum, and more increasingly so for larger datasets and more complex tasks.

**Model interpretation.** SHINE simultaneously learns the representations of nodes and hyperedges, which are then used to learn and inductively infer subgraph representations. This brings model interpretation advantages as it allows assessing pathways (hyperedges) correlations and reasoning multiple molecular functions mutually interacting and collectively contributing to the disease onset. We identify the top pathways that are enriched in different cancers using the attention weights learned for SHINE, as shown in Table 4. From the table, we see that many of the listed pathways reflect innate key events in the development of individual or multiple types of cancers, consistent with genetic and medical knowledge from wet lab (e.g., TNF/Stress Related Signaling [3]).

Table 3: Ablation Analysis: Held-out test set micro-F1 on real-world datasets. Standard deviations are provided from runs with 10 random seeds. SHINE significantly outperforms all the state-of-the-art comparison models. Best model in bold.

| Model                                    | DisGeNet Dataset Test Micro F1 | TCGA-MC3 Dataset Test Micro F1 |
|------------------------------------------|:------------------------------:|:------------------------------:|
| *Metrics*                                |                                |                                |
| HyperGCN (approx. hypergraph convolution) | $0.6638 \pm 0.0028$           | $0.4384 \pm 0.0095$           |
| HGNN (hypergraph convolution)            | $0.6809 \pm 0.0027$           | $0.4504 \pm 0.0042$           |
| HyperGAT (not strictly dual attention)   | $0.6495 \pm 0.0050$           | $0.4721 \pm 0.0032$           |
| SHINE without weighted subgraph attention | $0.6472 \pm 0.0053$          | $0.4388 \pm 0.0091$           |
| SHINE without hypergraph regularization  | $0.6829 \pm 0.0059$           | $0.5247 \pm 0.0048$           |
| SHINE                                    | $\mathbf{0.6955 \pm 0.0034}$  | $\mathbf{0.5319 \pm 0.0049}$  |

For breast cancer, TNF$\alpha$ is not only closely involved in its onset, progression and in metastasis formation, but also linked to therapy resistance [3]. Regarding the 4-1BB pathway, studies have suggested HER2/4-1BB bispecific molecule as a candidate of alternative therapeutic strategy to patients in HER2-positive breast cancer [4]. VIP/PACAP and their receptors have prominent roles in transactivation of the Epidermal growth factor (EGF) family and growth effects in breast cancer [5]. For lung cancer, the ErbB3 receptor recycling controlled by neuroregulin receptor degradation protein-1 is linked to lung cancer and small inhibitory RNA (siRNA) to ErbB3 shows promise as a therapeutic approach to treatment of lung adenocarcinoma [6]. Lung cancer is also modulated by multiple miRNAs interacting with the TFAP2 family [7]. For lower-grade gliomas, recent studies have reported the association between DNA demethylation and their malignant progressions [8]. Emerging evidence has also linked the citric acid (TCA) cycle for energy production to fuel the development of certain cancer types, especially those with deregulated oncogene and tumor suppressor expression [9]. For head and neck cancer, studies have reported a high percentage of cases with MECP2 copy-number gain and in combination with RAS mutation or amplification [10]. The apoptotic signaling and response pathways involving the mitochondrial pro-apoptotic protein SMAC/Diablo have also been suggested to regulate lipid synthesis that is essential for cancer growth and development [11].

Of note, the pathways listed in Table 4 for each cancer type play roles in different phases of cancer onset, growth or metastasis, and likely function together in tumorigenesis and progression, as discovered by SHINE. These analyses suggest that besides providing useful and discriminative features, SHINE integrates gene and pathway data to provide insights into functional and molecular mechanisms by linking together multiple pathways that may function together and contribute to cancer development and progression.

**Relevance and impact.** The techniques and results presented in the paper could apply to many diseases through informing genetic medicine practice. In these real-world applications, a subject's genetic profile may contain individual characterizing information. Thus, this work, or derivatives of it, should never be used in violation of an individual's privacy. For using individual level dataset such as the TCGA-MC3, the proper steps of IRB review of study and execution of data user agreement need to be properly completed prior to the study, such as done by this study.

It is important for the machine learning (ML) community to continue being informed about the problems arising in critical application domains such as healthcare and biomedicine that can guide model design. More specifically, explicitly treating hyperedges as first class citizens in the GNN modelling is important, since in this way hyperedges can be the subjects of notions of regularization or attention. This article demonstrated the feasibility to address those needs with our practical considerations of design and implementation choices by SHINE to advance modern genetic medicine study. We have demonstrated successful applications of SHINE on large-scale genetic medicine datasets, including the TCGA-MC3 dataset that is one of the largest NIH dbGaP datasets. Genetic medicine revolutionizes the practice of medicine in preventing, modifying and treating many diseases such as cardiovascular disease and cancer. In the future, as even larger genetic datasets will be collected through NIH programs such as All of Us and TopMed, we expect SHINE to be a useful tool in the quest of broadly advancing the knowledge on disease susceptibility.