# OpenReview forum: "SHINE: SubHypergraph Inductive Neural nEtwork"
_NeurIPS.cc/2022/Conference — NeurIPS 2022 Accept_

### Official Review · Reviewer_FtLi · 2022-06-27

**Rating:** 4
**Confidence:** 4
**Soundness:** 2 fair
**Presentation:** 3 good
**Contribution:** 2 fair

**Summary:**

Hypergraph neural networks can exploit multi-way connections in relational datasets but they are underexplored in domains such as genetic medicine. In this paper, a hypergraph attention-based message passing neural network is proposed for sub(hyper)graph-level tasks, e.g.,
* genes: nodes,
* pathways: hyperedges,
* patients: subgraphs,
* predict cancer type of patient: task.

Experiments on genetic medicine datasets demonstrate the effectiveness of the proposed method SHINE: SubHypergraph Inductive Neural nEtwork.

**Questions:**

1. How are the attention mechanisms in HyperGAT and SHINE different from existing attention mechanisms in heterogeneous graph neural networks with two different node types, e.g., treat genes and pathways as two different node types?
2. How does a simple feed-forward neural network with hypergraph regularisation and subgraph attention perform? This baseline is basically an ablated baseline with strongly dual attention message passing removed from SHINE and replaced with an MLP.
3. Is the strongly dual attention message passing scheme of SHINE an instantiation of the multiset formulation of "You are AllSet: A Multiset Function Framework for Hypergraph Neural Networks, In ICLR'22"?
4. What are the training times of all the methods (SHINE and all baselines)?
5. How sensitive are the hyperparameters (e.g., hidden dimensions) to the performance of SHINE? What is the optimal number of hidden layers in SHINE?

**Limitations:**

The authors have addressed the limitations and potential negative societal impacts adequately.

**Strengths And Weaknesses:**

**Originality**

Even though the paper explores an underexplored research topic in an interesting domain (subgraph representation learning for hypergraphs in genetic medicine), the methods proposed are incremental extensions of existing methods and not novel combinations of existing techniques.

Specifically in section 3.3., the ideas of hyperedge attention over nodes and node attention over hyperedges with parameter sharing are incremental extensions of well-known hypergraph attention networks.

By viewing the nodes and hyperedges as two types of vertices of a (bipartite) heterogeneous graph, the ideas of strongly dual attention mechanisms would be incremental extensions of existing attention-based methods for heterogeneous graphs, e.g., see "A Survey on Heterogeneous Graph Embedding: Methods, Techniques, Applications and Sources"

\
**Quality**

The authors have discussed interesting weaknesses of their work (in addition to highlighting the strengths).

Moreover, baseline comparison (Table 3), interpretability analysis (Table 4), and ablation study (Table 3 in supplementary) support the claims made in the paper empirically to an extent.

However, formalising the key differences with existing similar methods (e.g., HyperGAT in lines 159-169) and confirming the differences with convincing (synthetic/real-world) experiments, e.g., on a dataset chosen cleverly to show clear failure of HyperGAT but success of SHINE, would improve the paper's quality.

\
**Clarity**

The paper is well organised.

Details on datasets and hyperparameter tuning could help an expert to reproduce the results of the paper and build effective models (those with the best hyperparameters) from scratch.

A discussion on computational complexity and an algorithm/pseudo code would further enhance the clarity of the paper.

\
**Significance**

It is unclear from the paper why modelling genetic medicine datasets with hypergraphs, despite being a natural choice, is the best choice compared to straightforward alternatives.

More specifically, it is unclear why a (bipartite) heterogenous graph with genes: nodes of type 1, pathways: nodes of type 2, patients: (sub) heterogeneous graph would not be a reasonable choice.

The paper can be improved by positioning and comparing with set-based methods for exploiting hyperedges in hypergraphs, e.g., You are AllSet: A Multiset Function Framework for Hypergraph Neural Networks, In ICLR'22.

---

> ### Author Response · Authors · 2022-08-02
> **Response to Reviewer FtLi**
>
> Thank you for the detailed review and constructive suggestions!
> ### 1. Compare with SubGNN bipartite
> Thank you for suggesting the baseline and important reference “A Survey on Heterogeneous Graph Embedding: Methods, Techniques, Applications and Sources”. We will properly cite the reference. We have conducted the suggested experiment, and will add the following results. **New Results** SubGNN bipartite has held-out test set micro-F1 of 0.6137 $\pm$ 0.0097 on the DisGeNet dataset, having non-overlapping standard deviation intervals, in fact wide separation, with the results from SHINE. The experiment on the TCGA-MC3 dataset has taken over a couple of days and has not finished yet, but from what has come out, we are confident that the comparison will be similar as the case of the DisGeNet dataset. We will update the results to the revision when the full SubGNN experiment on the TCGA-MC3 dataset has finished.
> ### 2. MLP replacing dual attention ablation
> Thank you for suggesting the important ablation. We have conducted the suggested study and will add the following results. **New Results** MLP replacing dual attention message passing has obtained held-out test set micro-F1 of 0.6331 $\pm$ 0.0056 on DisGeNet dataset and 0.4249 $\pm$ 0.0165 on TCGA-MC3 dataset, both having non-overlapping standard deviation intervals with those from SHINE, separated by a large margin. We add the following to Discussion. “This has suggested that although MLP has been frequently used to approximate a target function, in the setting of large hypergraph (e.g., both hypergraphs have numerous thousand-nodes hyperedges), it can still be quite challenging to approximate an ideal target function and explicit dual attention formulation wins out.”
> ### 3. Differences with HyperGAT, and set-based methods AllSet
> Great point and suggestions! Key differences with HyperGAT: strongly dual attention (see next paragraph) and hypergraph regularization. HyperGAT has no hypergraph regularization that nodes with similar context of pathways (similar molecular functions) should have similar representations. So HyperGAT has no built-in measures to prevent or discourage genes belonging to the same functional class (e.g., promoting immune reactions) from having drastically different representations (e.g., opposite directions), a phenomenon that will pose interpretation difficulty.
>
> We will properly cite the seminal work AllSet, which subsumes HyperGAT as a special case. Strongly dual attention explores the hypergraph propagation from a different angle than both HyperGAT and AllSet. Compared to HyperGAT, strongly dual attention has the calculation of hyperedge’s and node’s attentions share the same underlying dual-attention matrix as shown in Fig. 1 (b). Such parameter sharing is meant to preserve the interchangeable nature of hypergraph’s nodes and hyperedges, and guarantee that (H*)* = H. When analogously applied to AllSet, strongly dual attention means to explore attention context sharing between fV->E and fE->V and will be a direction for future work.
>
> **New Results** We followed your suggestions of AllSetTransformer and AllDeepSets as baselines, and will add the following results and accompanying discussions. Their held-out test set micro-F1 are AllDeepSets: 0.6309 $\pm$ 0.0147 (DisGeNet), 0.4324 $\pm$ 0.0220 (TCGA-MC3); AllSetTransformer: 0.6355 $\pm$ 0.0160 (DisGeNet), 0.4904 $\pm$ 0.0158 (TCGA-MC3). The results from both AllDeepSets and AllSetTransformer have non-overlapping standard deviation intervals, in fact wide separation, with their counterparts from SHINE. These results echo with our observations that strongly dual attention explores the hypergraph propagation from a different angle than both AllDeepSets and AllSetTransformer, and suggest that effectively combining both angles could be an interesting future direction.
>
> ### 4. Hidden dimensions sensitivity, optimal K, hyperparameter details
> We will include a sensitivity analysis of hyperparameters in the supplement. In general, the performance is less sensitive to the hidden dimensions when it is at sufficiently big (300-500), with <0.05 change in micro-F1 score. Smaller hidden dimensions (100-200) can lead to >0.05 micro-F1 drop, likely due to insufficient representation power.
>
> We clarify that we varied the number K of layers from 1 to 4, and found that 2 layers to give the best results for SHINE. To better help an expert to reproduce the results of the paper and build effective models from scratch, we will add the download links from the mentioned data sources, and provide detailed preprocessing steps to create the datasets we used. We will also provide the best hyperparameters for the models in addition to the number K of layers.

---

> > ### Author Response · Authors · 2022-08-07
> > **Continued Response to Reviewer FtLi**
> >
> > ### 5. Computational complexity, training times, algorithm/pseudo code
> > Thank you for the suggestion. The complexity of SHINE scales as the following factors grow: the numbers of layers and nodes, the number and size of hyperedges, the size of hidden dimensions, and finally the number and size of subhypergraphs.
> > We will include the training times in the supplement. For a high level ballpark, for example on V100 GPU on the TCGA-MC3 dataset: MLP ~5min, HyperGCN ~7min, AllSetTransformer ~20min, AllDeepSet ~20min, SHINE ~30min, HGNN ~30min, HyperGAT ~30min, SubGNN >1day (excluding prebuild time).

---

> > > ### Author Response · Authors · 2022-08-09
> > > **Follow up with Reviewer FtLi**
> > >
> > > Dear Reviewer FtLi,
> > >
> > > Thank you for your constructive feedbacks and suggestions again! We greatly appreciate the additional rigorous ablation studies and state-of-the-art baselines (e.g., AllSetTransformer and AllDeepSets) that you suggested, and our new results have further strengthened the paper. We also highly value your comprehensive suggestions such as on hyperparameter sensitivity and computational complexity, addressing which have further increased the technical value of our paper. We hope that our responses have addressed your concerns and questions appropriately, and that you will consider increasing your score.
> > >
> > > Best Regards,
> > >
> > > Paper1592 Authors

---

### Official Review · Reviewer_noVi · 2022-07-08

**Rating:** 4
**Confidence:** 4
**Soundness:** 3 good
**Presentation:** 3 good
**Contribution:** 2 fair

**Summary:**

This paper suggests sub-hypergraph representation learning, a niche problem related to subgraph and hypergraph learning. To tackle this problem, the authors propose the SHINE (SubHypergraph Inductive Neural nEtwork) model, which consists of three modules: strongly dual attention message passing, hypergraph regularization, and weighted subgraph attention. Experiments on two real-world datasets demonstrate the superiority of SHINE on performance (against baselines including GNNs for hypergraphs) and interpretation (using attention).

**Questions:**

My questions are summarized in the weaknesses section.


**Limitations:**

The authors do not address the potential negative societal impact of their work. This paper targets a high-level machine learning problem called sub-hypergraph representation learning; however, all datasets are related to a particular area, genes, pathways, and diseases. There could be a potential societal impact that should be considered in real-world applications in this area (e.g., privacy). It would be nice if the authors addressed this point.


**Strengths And Weaknesses:**

## Strengths

The authors present a novel and niche problem of sub-hypergraph representation learning which has not been explored in the GNN community. A specific example (cancer patients as subgraphs of genes in hypergraphs) can be a practical application for this task. The performance improvement by the authors’ approach is significant.

## Weaknesses

However, I think this paper is not ready for publication for the following reasons.

First, the technical novelty of SHINE is limited. This model consists of several parts, and each of them is a slightly modified version of existing approaches. Using the attention to both nodes and (hyper) edges is presented in HyperGAT, and the authors are aware of it. Nevertheless, the idea of (strongly) dual attention is aligned with the dual form of hypergraphs, and I can see the novelty of this paper here. However, explicit regularization by Laplacian (Hypergraph regularization) [1] and pooling by attention weights (Weighted Subgraph Attention) [2] are well-known methods in using GNNs. In this case, SHINE's novelty is limited to Strongly Dual Attention, and a more detailed analysis of this part is required.

Second, related to the first paragraph, there are no rigorous ablation studies on the architecture. As many submodules make up the model, it is necessary to study where the performance gain comes from. In the supplementary material, only the ablation study on hypergraph regularization is presented, and the study on dual attention message passing is presented by comparison with HyperGAT. However, there are other differences in attention forms between HyperGAT and SHINE, and comparing these two does not provide a fully controlled ablation study of dual attention message passing. I recommend authors retain all parts except parameter-sharing in the attention. In addition, the performance comparison between SHINE with/without WSA and other GNNs with/without WSA also should be presented.

Third, it is skeptical that model interpretation by attention is an exclusive strength of SHINE. There are learned attentions in other attentional models like HyperGAT. Can these models provide interpretation at the same level as SHINE? Can you compare interpretations between models? Does SHINE give more precise explanations than other models?

Lastly, there are missing baselines for subgraph classification; in particular, the SubGNN can be a strong baseline. Of course, SubGNN is not designed for hypergraphs, but it is straightforward to create graphs from hypergraphs such as clique expansion. The transformation from hypergraphs to graphs is done only once before training; thus, it has a low overhead. Comparing SHINE and GNNs-for-subgraphs can justify that these specific problems in this work should be represented as a hypergraph.

## References

- [1] Learning with Hypergraphs: Clustering, Classification, and Embedding
- [2] GATED GRAPH SEQUENCE NEURAL NETWORKS

---

> ### Author Response · Authors · 2022-08-02
> **Response to Reviewer noVi**
>
> Thank you for the detailed review and constructive suggestions!
>
> ### 1. More analysis of Strongly Dual Attention; compare with SubGNN; more rigorous ablation studies on the architecture
> Thank you for suggesting more analysis of Strongly Dual Attention and for suggesting SubGNN as a baseline to justify the hypergraph representations of specific problems in this work. As Reviewer tNZm also pointed out, the importance of strong duality would also follow naturally from such comparison. **New Results** We have conducted the suggested experiment, and will add the following results. SubGNN bipartite has held-out test set micro-F1 of 0.6137 $\pm$ 0.0097 on the DisGeNet dataset, having non-overlapping standard deviation intervals, in fact wide separation, with the results from SHINE. The experiment on the TCGA-MC3 dataset has taken over a couple of days and has not finished yet, but from what has come out, we are confident that the comparison will be similar as the case of the DisGeNet dataset. We will update the results to the revision when the full SubGNN experiment on the TCGA-MC3 dataset has finished. We tried SubGNN clique expansion. However, the program was killed after exhausting all 1TB RAM on our server, due to large hypergraph and hyperedges.
>
> In addition, in Supplement Table 3, the comparison between “HyperGAT (not strictly dual attention)” and “SHINE without hypergraph regularization” is a direct ablation comparison of strongly dual attention where the difference is whether or not to retain parameter-sharing for hyperedge and node attention. As we mentioned in the section of WSA, HyperGAT does not directly support subgraph inferencing, and we added our WSA module to those models for subgraph inferencing. As can be seen, “SHINE without hypergraph regularization” clearly outperforms “HyperGAT (not strictly dual attention)” demonstrating the utility of strongly dual attention.
>
> **New Results** For more rigorous ablation studies on the architecture, we have followed you and the other Reviewers’ suggestions and added **5 state-of-the-art baselines and ablation studies**, including SubGNN bipartite (Reviewers tNZm, noVi, FtLi), weighted subgraph attention (WSA) ablation (Reviewers tNZm, noVi), MLP replacing dual attention ablation (Reviewer FtLi), AllSetTransformer (Reviewer FtLi), AllDeepSets (Reviewer FtLi). Please refer to Reviewer specific responses for the newly added results and accompanying analysis, which collectively have further strengthened our paper.
>
> ### 2. WSA ablation study
> Thank you for suggesting this important ablation case and suggesting the important citation [2] GATED GRAPH SEQUENCE NEURAL NETWORKS. We will properly cite it and will add the WSA ablation results where in general models without WSA show clear performance drop. **New Results** In particular, SHINE without WSA has obtained held-out test set micro-F1 of 0.6472 $\pm$ 0.0053  on DisGeNet dataset and 0.4388 $\pm$ 0.0091 on TCGA-MC3 dataset, both having non-overlapping standard deviation intervals with their counterparts from SHINE, separated by a wide margin. We add the following to the Discussion. “We also notice that the performance drop due to WSA ablation on the TCGA-MC3 dataset is larger than that on the DisGeNet dataset. This is consistent with the fact that the TCGA-MC3 dataset has denser hypergraph and larger subgraphs than the DisGeNet dataset. This is also consistent with the fact that differentiating among cancer subtypes is a more complex and nuanced task than differentiating among disease categories. These observations collectively argue for the benefits of weighted subgraph attention over direct aggregation such as sum, and more increasingly so for larger datasets and more complex tasks.”
>
> ### 3. Model interpretation by attention as strength
> We clarify that we did not mean to characterize that model interpretation by attention is an exclusive strength of SHINE. As the Reviewer correctly pointed out, other attention models such as HyperGAT can also provide attention based model explanation. Currently few quantitative interpretation comparison methods (see Yuan et al.; reference below) are applicable to compare the genetic pathways interpretation, as genetic pathways correspond to molecular functions and it is hard to quantify a molecular function’s value to a living organism. However, we do note that HyperGAT has no hypergraph regularization that nodes with similar context of pathways (molecular functions) should have similar representations. Thus HyperGAT has no built-in measures to prevent or discourage genes belonging to the same functional class (e.g., promoting immune reactions) from having drastically different representations (e.g., opposite directions), a phenomenon that will pose interpretation difficulty.
>
> _Yuan H, Yu H, Gui S, Ji S. Explainability in graph neural networks: A taxonomic survey. arXiv preprint arXiv:2012.15445. 2020 Dec 31._

---

> > ### Author Response · Authors · 2022-08-03
> > **Continued Response to Reviewer noVi**
> >
> > ### 4. Potential negative societal impact
> > Thank you for suggesting the discussions on potential negative societal impact and we will add the following paragraph to Discussion.
> > “The techniques and results presented in the paper could apply to many diseases through informing genetic medicine practice. In these real-world applications, subject’s genetic profile may contain individual characterizing information. Thus, this work, or derivatives of it, should never be used in violation of individual’s privacy. For using individual level dataset such as the TCGA-MC3, the proper steps of IRB review of study and execution of data user agreement need to be properly completed prior to the study, such as done by this study.”

---

> > > ### Author Response · Authors · 2022-08-09
> > > **Follow up with Reviewer noVi**
> > >
> > > Dear Reviewer noVi,
> > >
> > > Thank you again for your constructive comments and suggestions! They help articulate our contribution, lead to more balanced discussions (e.g., interpretation and societal impact), and result in more rigorous ablation studies that have further strengthened our paper. We hope that our responses have properly addressed your concerns and that you will consider increasing your score.
> > >
> > > Best Regards,
> > >
> > > Paper1592 Authors

---

### Official Review · Reviewer_tNZm · 2022-07-11

**Rating:** 7
**Confidence:** 4
**Soundness:** 3 good
**Presentation:** 3 good
**Contribution:** 3 good

**Summary:**

The authors propose a GNN approach to learn embeddings of sub-hyper-graphs. The approach has an explicit treatment of hyper edges, e.g., it does not resort to clique expansion, and makes use of a regulariser based on the hyper graph laplacian.  The application chosen is that of disease prediction for patients (modelled as sub hyper graphs) given a pathway network (modelled as a hyper graph with genes as nodes and sets/pathways as hyper edges).

**Questions:**

Additional experiments to quantify the   relative importance of the various ideas introduced would strengthen the paper.
In particular consider offering support to reply to the points 1, 2 and 3 raised as negative elements in the Strengths And Weaknesses section.

**Limitations:**

yes.

**Strengths And Weaknesses:**

Pos
Explicitly treating hyper edges as first class citizens in the GNN modelling is of interest, since in this was hyper edges can be the subjects of notions of regularisation or attention.

Neg
The relative importance of the various ideas introduced is not clear, i.e., a better experimental design with clearer baselines and an ablation study is warranted. More specifically:

1. is the regularisation effective or of importance? (ablation case)

2. is the proposed architecture much different from using a standard graph neural network with attention on a pre-processed hyper graph? In particular the pre-processing could consist in representing an hyper graph as a bi-partite graph where hyper edges are materialised as the nodes of one part and the genes are the nodes of the other part.  (experiment)
Note that the whole discussion regarding strong duality would follow automatically in this case.

3. is the introduction of WSA (the weighted  subgraph attention) needed? what happens if we replace the whole subgraph treatment by Mji directly? I.e., what if we consider a subgraph simply the sum of the nodes (genes) that are of interest (with mutations) for each patient (subgraph), that is, we could learn directly the embedding of the nodes/hyperedges for the classification task when they are simply summed up for each patient. (experiment/ablation)

---

> ### Author Response · Authors · 2022-08-02
> **Response to Reviewer tNZm**
>
> Thank you for the detailed review and constructive suggestions!
> ### 1. Is the regularisation effective or of importance
> Thank you for pointing out the importance of the ablation case of the hypergraph regularization. We clarify that our Supplemental Table 3 has presented this case: SHINE without hypergraph regularization has held-out test set micro-F1 of 0.6829 $\pm$ 0.0059 on the DisGeNet dataset and 0.5247 $\pm$ 0.0048 on the TCGA-MC3 dataset, both having non-overlapping standard deviation intervals with their counterparts from SHINE. This suggests that that adding hypergraph regularization further improves performance. We will add the following discussion to provide intuition. “It is known that Graph Convolutional Network suffers from oversmoothing when the number of layers increases, as increasingly globally uniform representation of nodes may be developed. On the other hand, attention could limit this phenomenon by limiting to a restricted set of nodes. The effect of hypergraph regularization, while also smoothing, happens on a local scale as part of a direct optimization objective and does not accumulate with increasing number of layers. Such decoupling between attention and local smoothing allows SHINE to better explore the optimization landscape.” Given the shared interest on ablation studies, we will move our ablation analysis discussions to the main paper, integrating additional ablation cases suggested by you and other Reviewers.
>
> ### 2. Adding experiment on hypergraph as a bi-partite graph
> Thank you for suggesting this important experiment. We agree that the importance of strong duality would follow naturally from comparing SHINE with hypergraph as a bi-partite graph where hyperedges are materialized as the nodes of one part and the genes are the nodes of the other part. We have conducted the suggested experiment based on the implementation of SubGNN, and will add the following results. **New Results** SubGNN bipartite has held-out test set micro-F1 of 0.6137 $\pm$ 0.0097 on the DisGeNet dataset, having non-overlapping standard deviation intervals, in fact wide separation, with the results from SHINE. The experiment on the TCGA-MC3 dataset has taken over a couple of days and has not finished yet, but from what has come out, we are confident that the comparison will be similar as the case of the DisGeNet dataset. We will update the results to the revision when the full SubGNN experiment on the TCGA-MC3 dataset has finished.
>
> ### 3. WSA (the weighted subgraph attention) ablation study, I.e., what if we consider a subgraph simply the sum of the nodes (genes) that are of interest (with mutations) for each patient (subgraph)
> Thank you for suggesting this important ablation case. We have followed your suggestion on assessing the impact of the introduction of WSA (the weighted subgraph attention) and will add the following results. **New Results** SHINE without WSA has obtained held-out test set micro-F1 of 0.6472 $\pm$ 0.0053 on the DisGeNet dataset and 0.4388 $\pm$ 0.0091 on the TCGA-MC3 dataset, both having non-overlapping standard deviation intervals with their counterparts from SHINE, separated by a wide margin. We add the following to the Discussion. “We also notice that the performance drop due to WSA ablation on the TCGA-MC3 dataset is larger than that on the DisGeNet dataset. This is consistent with the fact that the TCGA-MC3 dataset has denser hypergraph and larger subgraphs than the DisGeNet dataset. This is also consistent with the fact that differentiating among cancer subtypes is a more complex and nuanced task than differentiating among disease categories. These observations collectively argue for the benefits of weighted subgraph attention over direct aggregation such as sum, and more increasingly so for larger datasets and more complex tasks.”
> ### 4. Additional experiments to quantify the relative importance of the various ideas introduced would strengthen the paper.
> **New Results** Many thanks to the ablation cases suggested by you (e.g., points 1, 2, 3) and the other Reviewers, we now have a further expanded and comprehensive experiments and results by adding **five additional state-of-the-art baselines and ablation studies**, including SubGNN bipartite (Reviewers tNZm, noVi, FtLi), WSA ablation (Reviewers tNZm, noVi), hypergraph regulation ablation (Reviewer tNZm), MLP replacing dual attention ablation (Reviewer FtLi), AllSetTransformer (Reviewer FtLi), AllDeepSets (Reviewer FtLi). Please refer to Reviewer specific responses for these newly added results and accompanying analysis, which have indeed further strengthened the paper, thank you for the constructive suggestions!

---

> > ### Author Response · Authors · 2022-08-09
> > **Follow up with Reviewer tNZm**
> >
> > Dear Reviewer tNZm,
> >
> > Thank you again for your constructive comments and suggestions! We greatly appreciate your suggested baselines/ablations, and our new results have further strengthened the paper. We hope that our responses have properly addressed your questions and concerns and that you will consider increasing your rating.
> >
> > Best Regards,
> >
> > Paper1592 Authors

---

### Official Review · Reviewer_CYi2 · 2022-07-12

**Rating:** 6
**Confidence:** 4
**Soundness:** 3 good
**Presentation:** 3 good
**Contribution:** 2 fair

**Summary:**

The paper proposes a hypergraph neural network model exploiting a double attention mechanism in the message passing scheme. The overall architecture is designed to process sub-hypergraphs once representations are computed for the nodes and edges. The learning objective includes a regularization term based on the hypergraph laplacian.
The proposed model is evaluated on disease classification based on gene-genetic pathways data, showing higher F1 values with respect to a set of competitors.
Finally, due to the attention mechanism intepretions in terms of gene pathways (hyperedges) can be derived from the model outputs.

**Questions:**

How performances are affected by the number K of layers?

**Limitations:**

The authors discuss some limitations that need futher work in section of the paper both for the model architecture and the specific application considered for the evaluation.
As listed in the weaknesses, a more general description (not tailored for the considered task in genetic medicine) would have improved the presentation of the proposed hypergraph neural network architecture.

**Strengths And Weaknesses:**

Strengths
- the model incorporates a dual attention mechanism applied to nodes and hyperedges respectively, exploiting the same attention context vector. This desing choice is claimed to prevent overfitting reducing the number of parameters
- the architecture includes an attention module to derive subgraph representations. This scheme allows the application of the mode in a inductive setting
- the model compares favourably with respect to the considered competitors on two benchmarks in genetic medicine

Weaknesses
- The presentation of the model is tightly interconnected with the proposed application in genetic medicine, making it appear less general
- Given the focus on bioinformatics, the paper is hard to follow for readers not completely familar with this topic.
- The effect of the number K of layers is not investigated (maybe I missed it, but the number of layers used in the experiments is not reported). It is known that (Convolutional) Graph Neural Network suffer from oversmoothing when the depth of the network increases. This may hinder the results in some applications since uniform representation of nodes are developed. Attention may perhaps limit this phenomenon, but on the other side node/edge regularization may produce add a related effect. The ablation study show a positive effect of regularization, but some discussion/analysis should be provided.

---

> ### Author Response · Authors · 2022-08-02
> **Response to Reviewer CYi2**
>
> Thank you for the detailed review and constructive suggestions!
> ### Presentation of model tightly interconnected with genetic medicine, a more general description can improve the presentation
> Thank you for suggesting a better presentation strategy for the model and will conduct an overhaul to make the model description more general. For example, in the description of hypergraph learning, we will consistently refer to hyperedges instead of genetic pathways and leave the interconnection with genetic medicine starting from the description of experiments.
> We will also add the following discussion to explain why the field of genetic medicine is more general than it appears, and in fact, impacts the whole field of medicine. “The field of genetic medicine encompasses areas of molecular biology and clinical phenotyping to explore new relationships between disease susceptibility and human genetics. Though appearing as a single field, it revolutionizes the practice of medicine in preventing, modifying and treating many diseases such as hypertension, obesity, diabetes, arthrosclerosis, and cancer (see Green et al; reference below). Our carefully chosen experimental datasets simultaneously considers the availability of large public data and the demonstration of general medicine applications: TCGA-MC3 being across all major cancer types, and DisGeNet being across all major disease categories.”
>
> _Green, Eric D., et al. "Strategic vision for improving human health at The Forefront of Genomics." Nature 586.7831 (2020): 683-692._
> ### Making paper easier to follow for non-bioinformatics readers
> Thank you for suggesting better accessibility of the paper by general audience. We add to Introduction the following elaboration and example illustrating some intuitions behind the proposed model, in which we tried to use descriptive language instead of jargons for easier readability by general audience. We will similarly overhaul other places in the paper with more descriptive and accessible languages when discussing genetic medicine applications.
>
> “Genetic pathways are a valuable tool to assist in representing, understanding, and analyzing the complex interactions between molecular functions. The pathways contain multiple genes (can be modeled using hyperedges) and correspond to genetic functions including regulations, genetic signaling, and metabolic interactions. They have a wide range of applications including predicting cellular activity and inferring disease types and status (see Alon 2019; reference below). For a simplified and illustrative example, a signaling pathway p1 (having 20 genes) sensing the environment may govern (the governing function embodied as a pathway p2 having 15 genes) the expression of transcription factors in another signaling pathway p3 (having 23 genes), which then controls (the controlling function embodied as a pathway p4 having 34 genes) the expression of proteins that play roles as enzymes in a metabolic pathway p5 (having 57 genes). In general, there will be partial overlap between pathways p1 and p2, p2 and p3, p3 and p4, p4 and p5, and other potential partial overlaps corresponding to partial overlaps between their corresponding hyperedges.”
>
> _Alon U. “An introduction to systems biology: design principles of biological circuits”. CRC press; 2019._
>
> We also add to Supplement more background and context information for general audience, for example, explaining what the datasets look like, as follows:
> “The genetic variants are stored in a specially formatted file. A row in the file specifies a particular variant (e.g., Single Nucleotide Polymorphism or insertion/deletion), its chromosomal location, and what proportion of the sequencing reads covering that chromosomal location have that variant, among other characteristics.”
>
> ### Effect of the number K of layers
> Thank you for suggesting discussion/analysis on K. We add the clarification that we varied the number K of layers from 1 to 4, and found that 2 strongly dual attention layers (followed by a weighted subgraph attention layer) gave the best results for SHINE. For the original and added baselines/ablations (e.g., SubGNN, AllSetTransformer, AllDeepSet) models, we followed their respective papers in selecting the number K of layers, e.g., also varying K from 1 to 4 for SubGNN.
>
> We completely agree with the Reviewer’s assessment and add the following to Discussion. “It is known that Graph Convolutional Network suffers from oversmoothing when the number of layers increases, as increasingly globally uniform representation of nodes may be developed. On the other hand, attention could limit this phenomenon by limiting to a restricted set of nodes. The effect of hypergraph regularization, while also smoothing, happens on a local scale as part of a direct optimization objective and does not accumulate with increasing number of layers. Such decoupling between attention and local smoothing allows SHINE to better explore the optimization landscape.”

---

> > ### Author Response · Authors · 2022-08-09
> > **Follow up with Reviewer CYi2**
> >
> > Dear Reviewer CYi2,
> >
> > Thank you for your valuable comments and questions again! They have helped both clarify the technical aspects and improve the writing of our paper. We hope that our responses have properly addressed your questions and concerns and that you will consider increasing your rating.
> >
> > Best Regards,
> >
> > Paper1592 Authors

---

### Author Response · Authors · 2022-08-02
**Multiple new experiments added and thank you for your constructive comments!**

We thank all Reviewers for their time and feedback! We were glad to see that our work was in general positively received, and that Reviewers commented that “Explicitly treating hyper edges as first class citizens in the GNN modelling is of interest, since in this was hyper edges can be the subjects of notions of regularisation or attention” (Reviewer tNZm), “The authors present a novel and niche problem of sub-hypergraph representation learning which has not been explored in the GNN community. … (strongly) dual attention is aligned with the dual form of hypergraphs, and I can see the novelty of this paper here” (Reviewer noVi), “the model compares favorably with respect to the considered competitors” (Reviewer CYi2), “The paper is well organised” (Reviewer FtLi). We also appreciate their questions, comments, and suggestions.

**New Results** We have followed the Reviewers’ suggestions and conducted further experiments on **five additional state-of-the-art baselines and ablation studies**, including SubGNN bipartite (Reviewers tNZm, noVi, FtLi), weighted subgraph attention (WSA) ablation (Reviewers tNZm, noVi), MLP replacing dual attention ablation (Reviewer FtLi), AllSetTransformer (Reviewer FtLi), AllDeepSets (Reviewer FtLi). Please refer to Reviewer specific responses for the newly added results and accompanying analysis, which collectively have further strengthened our paper.

We provide answers to the main points raised by each Reviewer below, and outline changes that we plan to address in a potential revision. Please feel free to follow up with us! We very much welcome any feedback that can further strengthen the paper.

---

### Author Response · Authors · 2022-08-07
**Navigation summary of new results**

### Navigation summary of new results
For easier navigation, we provide clickable links in this followup for our **five additional state-of-the-art baselines and ablation studies** per Reviewers' suggestions as follows, including SubGNN bipartite (Reviewers [tNZm](https://openreview.net/forum?id=IsHRUzXPqhI&noteId=soQcEa9QjBx) #2, [noVi](https://openreview.net/forum?id=IsHRUzXPqhI&noteId=xrsJ4_Q5L2s) #1, [FtLi](https://openreview.net/forum?id=IsHRUzXPqhI&noteId=ZREzLf4f6Ke) #1), weighted subgraph attention (WSA) ablation (Reviewers [tNZm](https://openreview.net/forum?id=IsHRUzXPqhI&noteId=soQcEa9QjBx) #3, [noVi](https://openreview.net/forum?id=IsHRUzXPqhI&noteId=xrsJ4_Q5L2s) #2), MLP replacing dual attention ablation (Reviewer [FtLi](https://openreview.net/forum?id=IsHRUzXPqhI&noteId=ZREzLf4f6Ke) #2), AllSetTransformer (Reviewer [FtLi](https://openreview.net/forum?id=IsHRUzXPqhI&noteId=ZREzLf4f6Ke) #3), AllDeepSets (Reviewer [FtLi](https://openreview.net/forum?id=IsHRUzXPqhI&noteId=ZREzLf4f6Ke) #3). These newly added results and accompanying analysis collectively have further strengthened our paper. Thank you for your constructive suggestions!

---

### Meta-Review · Area_Chair_eQ67 · 2022-08-30

**Recommendation:** Accept
**Confidence:** Less certain

**Metareview:**

The paper proposed a GNN that explicitly treats hyperedges, and makes use of strongly dual attention, hypergraph regularization, and weighted subgraph attention.  The proposed method shows better performance than existing baselines on two genetic medicine datasets.  Explainability is also demonstrated.

Reviewers originally raised many concerns on presentation (too specialized for the target application), lack of ablation (effectiveness of each proposed component is not clearly shown), novelty (combination of small modifications of existing methods), and explainability (existing methods can do the same).  The authors made an amazing job to address most of the concerns:  They reported additional ablation results and baseline results, and showed that the proposed method still performs better, and each proposed component plays a significant role.

Two reviewers have been convinced by the author's response, while the other two have not, insisting that the novelty issue remains, and with the limited novelty, more careful investigation is required for publication.

This is a borderline paper, and I recommend acceptance because I think adjusting existing methods to target applications is important research even if the modifications are small.  The proposed method significantly outperforms existing baselines (including the ones reviewers suggested), and the additional ablation study shows each of the proposed components is effective.

On the other hand, I also sympathize with the reviewers with negative evaluations on the following comments:

"formalising the key differences with existing similar methods (e.g., HyperGAT in lines 159-169) and confirming the differences with convincing (synthetic/real-world) experiments, e.g., on a dataset chosen cleverly to show clear failure of HyperGAT but success of SHINE, would improve the paper's quality."

"The paper can be strengthened by positioning strongly dual attention in SHINE with different attention mechanisms in heterogeneous graph neural network literature (some are listed below):

Heterogeneous Graph Attention Network, In WWW'19
HetGNN: Heterogeneous Graph Neural Network, In KDD'19
Metapath enhanced graph attention encoder for HINs representation learning, In BigData'19.
MAGNN: Metapath Aggregated Graph Neural Network for Heterogeneous Graph Embedding, In WWW'20
Heterogeneous Graph Transformer, In WWW'20.

There is no need to empirically compare and run them as baselines but explaining the key differences conceptually to make hypergraphs a more compelling choice for genetic medicine than heterogeneous graphs can strengthen the paper."

I hope the authors would make a bit more effort to incorporate these suggestions in the final version.



**Award:**

No

---

### Decision · Program_Chairs · 2022-09-14

Accept